

# A Hydrate Reservoir Renovation Device and Its Application in Nitrogen Bubble Fracturing

Jingsheng Lu [1,5,6], Yuanxin Yao [1,5,6], Dongliang Li[1,3,5,6*], Jinhai Yang[2], Deqing Liang[1,3,5,6], Yiqun Zhang[4,5], Decai Lin[1,3,5,6] Kunlin Ma[1,3,5,6]

[1]Key Laboratory of Gas Hydrate, Guangzhou Institute of Energy Conversion, Chinese Academy of Sciences, Guangzhou 510640, China;

[2]Hydrates, Flow Assurance & Phase Equilibria Research Group, Institute of GeoEnergy Engineering, Heriot-Watt University, Edinburgh EH14 4AS, UK;

[3] School of Energy Science and Engineering, University of Science and Technology of China, Guangzhou 510640, China;

[4]China University of Petroleum, Beijing 102249, China;

[5]State Key Laboratory of Natural Gas Hydrate, Beijing 100028, China;

[6]Guangdong Provincial Key Laboratory of New and Renewable Energy Research and Development, Guangzhou 510640, China;

*Correspondence to*: Dongliang LI (ldl@ms.giec.ac.cn)

**Abstract.** Natural gas hydrate (GH) is a significant potential energy source due to its large reserves, wide distribution, high energy density and low pollution. However, the gas production rate of past gas hydrate production tests is much lower than the requirement of commercial gas production. Reservoir stimulation technologies like hydraulic fracture provide one potential approach to enhance gas production from GH. The reservoir reformation behaviour of the hydrate-bearing sediments (HBS), particularly sediments with a high clay content, is a complex process during a hydraulic fracturing operation, which has been poorly understood and thus hardly predictable. This paper presents an experimental facility that was developed to analyze the hydraulic fracture mechanism in synthesized HBS. This facility can be used to form GH in sediments, conduct visual observation of hydraulic fracturing experiments, and measure the permeability of HBS under high pressure (up to 30 MPa) and low-temperature conditions (from 253.15 K to 323.15K). It is mainly composed of a pressure control and injection unit, a low temperature and cooling unit, a cavitation unit, a visual sapphire reactor, and a data acquisition and measurement unit. The hydraulic fracture module is consisting of a gas cylinder, fracturing pump, hopper, proppants warehouse and valves. The sapphire reservoir chamber is applied to observe and measure the fracture of HBS during hydraulic fracturing. The permeability test module is composed of a constant-flux pump and pressure sensors, which can evaluate the permeability performance before and after hydraulic fracture in HBS. The fundamental principles of this apparatus are discussed. Some tests were performed to verify hydraulic fracture tests and permeability tests could be practically applied in the HBS exploitation.

## 1 Introduction

Nature gas hydrate (GH) is an ice-like crystal substance, named fire in ice, which is formed by water and gas under low temperature and high-pressure conditions(Sloan and Koh, 2007). It has largely stored in the deep-water and permafrost



sediments(Boswell, 2009). GH has been considered as a potential low-carbon energy source in the 21$^{st}$ century. The methods of depressurization test(Tang et al., 2007), thermal simulation test(Wang et al., 2014), inhibitor injection test(Tohidi et al.,

2015), carbon dioxide replacement test(Boswell et al., 2017) and solid fluidization test(Zhou et al., 2018) are applied to GH production in the last score years ago. However, the production rate of methane in these tests cannot meet the commercial requirement, and the key factor of hydrate commercial production is daily production rates(Chen et al., 2022; Yamamoto et al., 2022). Thus, the stimulation technology of HBS should be considered to achieve an economically viable gas production rate from GH reservoirs(Wu et al., 2021).

Hydraulic fracturing is one of the useful stimulation technologies widely applied to the "shale gas revolution" in the last three decades, which is also investigated to enhance production technology for GH(Terzariol and Santamarina, 2021; Terzariol et al., 2017)(Figure 1). Few hydraulic fracture studies of HBS were reported recently(Sun et al., 2019; Shen et al., 2020; Yao et al., 2021; Sun et al., 2021; Zhong et al., 2020; Lv et al., 2021; Konno et al., 2016; Ito et al., 2008; Ma et al., 2022). One challenge is how to detect the fracturing ability and features of HBS under low temperature and high pressure conditions.

While, the weak cementation, low permeability, and high fine content behaviour of HBS may lead to sand production(Lu et al., 2019), wellbore collapse and formation instability(Wu et al., 2019; Li et al., 2016; Wu et al., 2023b) during the fracturing stimulation operation. Although the innovative experimental apparatus for sand production(Lu et al., 2021a, 2018), cavitating jet(Zhang et al., 2020), mechanical behaviour (Seol et al., 2019; Spangenberg et al., 2020; Li et al., 2019) and kinetics behaviour (Masoudi et al., 2019) of HBS were developed, it is still hard to evaluate the fracturing performance (like fracture

generation, growth and determination) in HBS. Meanwhile, proppants are widely applied in hydraulic facture fluids to increase the permeability of unconventional reservoirs(Ahmed Hafez Abdelaziz, 2020; Wang et al., 2023a, b). The performance of proppants in HBS also a key factor of the stimulation technology. Furthermore, the fractures of HBS may trigger submarine slope failure and seafloor destabilization during the GH natural dissolution by global warming and marine salinity changes(Hassanpouryouzband et al., 2020). It is significant to study the fracture initiation and propagation mechanism in HBS

and how the fractures respond to the changes in the sedimentary properties and temperature and pressure conditions during hydraulic stimulation and, exploitation as well as the natural dissolution process.



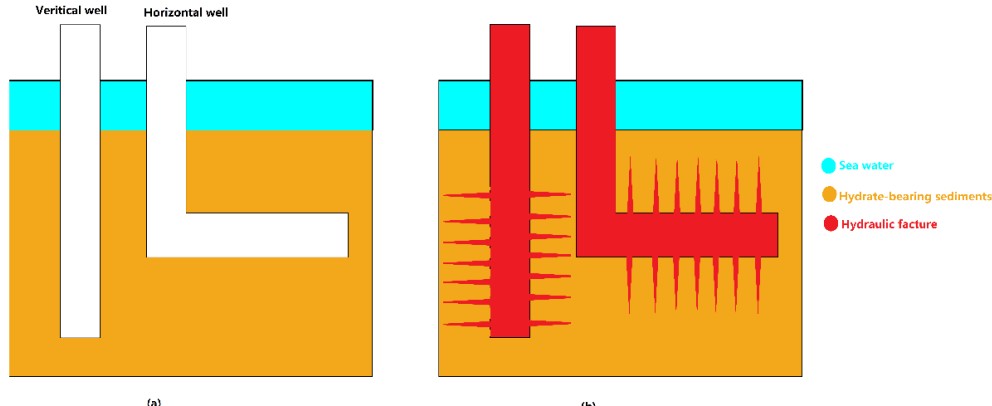

**Figure 1 Schematic diagrams of (a) hydrate-bearing sediment exploitation in a vertical well and a horizontal well (b) hydraulic facture in HBS**

However, the expensive cost of field tests and restriction of numerical simulation leads to the laboratory hydraulic fracture of HBS as the best option(Tang et al., 2007). To study the ability of stimulation using hydraulic facture in HBS, a novel experimental apparatus that consists of a set of hydraulic facture hydrate equipment was designed and developed. It was successfully used to study the ability and feature of the hydraulic fracture in HBS and the coupling effects of multi-field (thermos-hydro-mechanical-phase change) on GH exploitation under reservoir conditions.

## 2 Design focus

The marine HBS is usually buried in deep-water (1200 m) with high compaction stress (10-25 MPa), high pore pressure (10-20 MPa) and low temperature (275.15-288.15 K), so the effect of high crustal stress, high pressure and low temperature on hydraulic fracture could not be ignored during the stimulation process. Three key factors should be considered in the design: (1) the HBS formation,

(2) in-situ hydraulic fracture tests of HBS at high pressure and low temperature, and (3) fracture visualization of HBS under in-situ conditions. The schematic configuration of the designed apparatus, which is composed of a pressure control and injection unit, a low temperature and cooling unit, a cavitation unit, a visual sapphire reactor, a data acquisition and measurement unit, is shown in Figure 2.



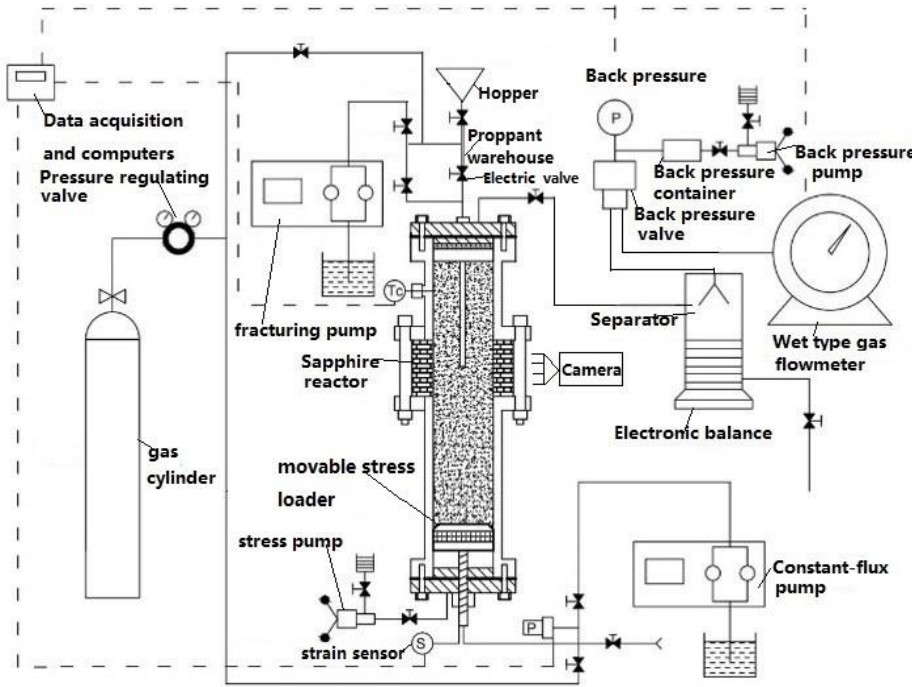

**Figure 2 Schematic diagram of equipment for HBS of hydraulic fracture**

## 2.1 Pressure and stress control system

In Figure 3 and Figure 4, the pressure and stress control system consists of four modules: the movable stress module (movable stress loader, stress pump and strain sensor), the hydraulic fracture module, the backpressure module (backpressure pump, backpressure container and backpressure valve), and pore pressure module (gas pressure and constant-flux pump). The pressure of the movable stress module, the hydraulic fracture module, the backpressure module, and the pore pressure module are provided by the stress pump (30 MPa), constant-flux pump (30 MPa), fracturing pump (30 MPa), backpressure pump (10 MPa) and methane gas (13 MPa), respectively. The automatic pressure relief valve is fixed to avoid pressure over the system limit. The strain sensor is assembled on a movable stress loader to measure the axial deformation (subsidence).



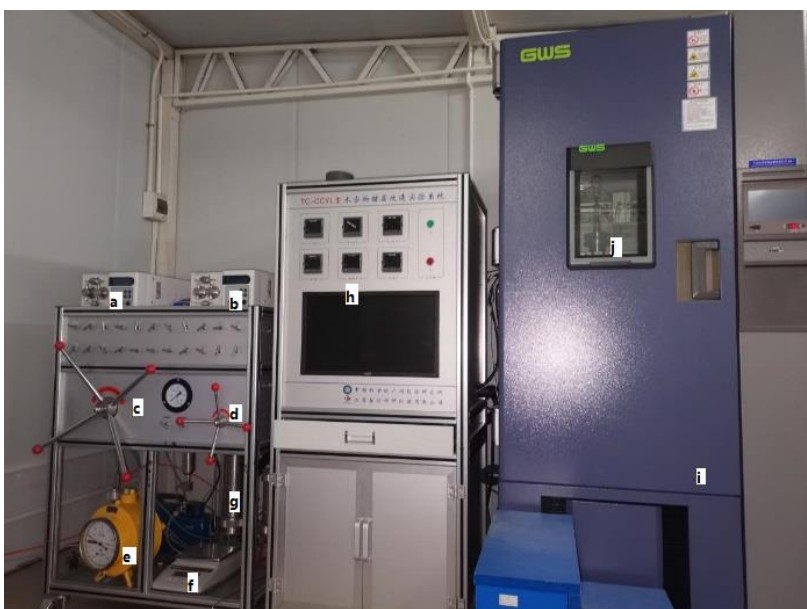

**Figure 3 The front view of the apparatus: (a) Fracturing pump (b) Pore pressure pump (c) Overlying stress (d) Back pressure pump (e) Wet type gas flowmeter (f) Electronic balance (g) Separator (h) Control cabinet and computer (i) Air bath (j) Visual window of air bath and sapphire reactor**

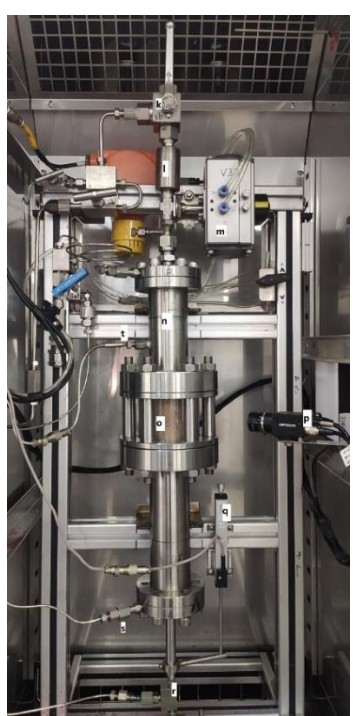

**Figure 4 The inside-view of the air bath:(k)Hopper valve; (l) Proppant warehouse; (m) Electric valve of proppant warehouse; (n)**
**Up chamber of the sapphire reactor; (o) Sapphire window of the sapphire reactor; (p) Camera; (q) Strain sensor; (r) Pore pressure valve and safety valve; (s) Stress valve; (t) Temperature sensor.**



## 2.2 Low temperature and air cooling system

To control the temperature in the reactor, the programmable air bath is applied (Figure 5). The programmable air bath is manufactured by Guangzhou-GWS Environmental Equipment Co., Ltd, which can provide a temperature range from 253.15
K to 323.15 K, and the accuracy is ±0.5 K.

It applied the 380 V voltage for cooling power. The visual window and inside light of the programmable air bath are applied to the visual reactor by eye and camera. There is a temperature sensor (PT-100, the accuracy is ±0.1 K) arranged in the middle of the reactor (Figure 4), which can collect the reactor temperature in real-time.

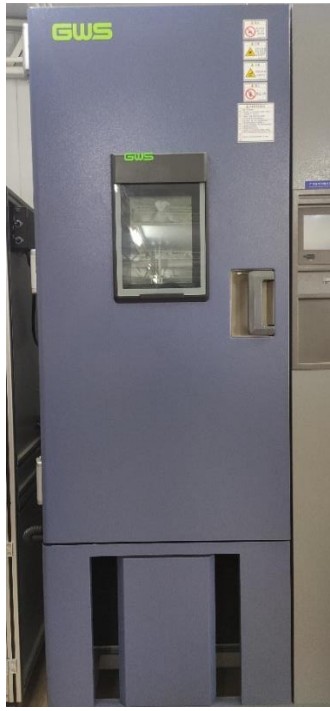

**Figure 5 The programmable air bath.**

## 2.3 Hydraulic fracture, permeability test and production system

The hydraulic fracture module consists of the gas cylinder, fracturing pump, hopper, proppants warehouse and valves. After adding the proppants into the warehouse through the hopper, the hydraulic fracture pressure increased with coloured water (fracturing fluid) and $N_2$ gas by the fracturing pump and gas cylinder, respectively. The high-pressure hydraulic fracturing fluid
with proppants flows directly through the pipe (Φ 8 mm) into the visual sapphire reactor when the electric valve of the proppants warehouse opens.

The permeability test module is composed of the constant-flux pump and pressure sensors. It determines the permeability of HBS before and after hydraulic fracturing through Darcy's law(Wu et al., 2023a; Lu et al., 2021b).



The production module is constituted of a backpressure module, separator, electronic balance and gas flowmeter. It is applied
to test the production capacity of HBS after the hydraulic fracture.

### 2.4 Visual sapphire reactor

The visual sapphire reactor (Figure 4) is divided into three parts of the up chamber (Φ 40 mm × 140 mm, 125 ml), visual
window chamber and down chamber (Φ 40 mm ×200 mm, 250 ml). The body material of the up and down chamber is stainless
steel 316L with an O ring seal, which can tolerate 20 MPa. The sapphire hollow cylinder (Φ 40 mm ×60 mm, 75 ml) is applied
to the visual window chamber for eye and camera monitoring.

### 2.5 Data acquisition and measurement control system

The digital acquisition and control card are applied to ensure real-time data acquisition by MOX C168H. Through the
control cabinet (Figure 6), the hydrate fracturing pressure, pore injection pressure (bottom pressure), pore pressure (up
pressure), production pressure (up pressure), temperature and movable stress can be displayed and collected. The gas
measuring equipment is the BSD05 wet flow meter for gas monitoring (measuring range 12.5L /min, ±1%) by Krom
Co.Ltd. The pressure sensor is manufactured by TraFag.Co.Ltd with a range of 30 MPa and an accuracy of 0.1%F.S.

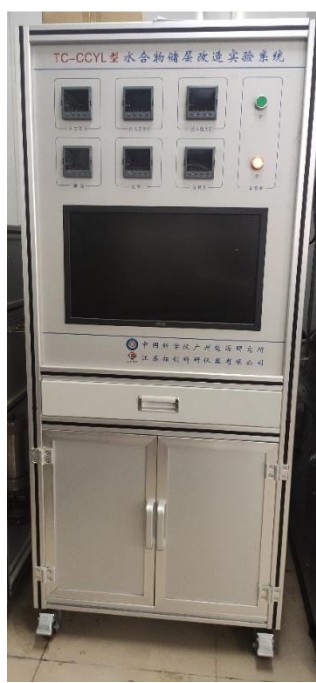

**Figure 6 Control cabinet**

The experimental process and the related control parameters of equipment are controlled by VB-compiled experiment
measurement control software. The data of real-time acquisition are reactor chamber internal pressure, production
pressure, injection pressure, movable pressure and other parameters. The experimental data can be output in Excel form.



The software can set and control the electric valve.

## 3. EXPERIMENTAL PROCESS AND RESULTS

### 3.1 Hydrate formation

The specific experimental process is as follows:

(1) Sample formation:

First, the hole of the hydraulic jet pipe is coated with a thin filter paper, which is to prevent sand from entering the hydraulic jet pipe. Then sediments with a certain moisture content are put into the reactor. For the compaction of the sediments sample, 1 MPa stress is applied by the movable stress loader for 1 min. After vacuuming the system for 5 min, the methane gas was

injected into the reactor with a stress loader (effective stress of no more than 1 MPa). Finally, the pore pressure and stress reached equilibrium at 10 MPa and 11 MPa, respectively. After settling down at 293.15 K for 24 h (methane, water and sediments fully mixed), the temperature of the reactor was cooled to 274.15 K. The pressure in the reactor was gradually balanced at about 72 to 144 h, while the hydrate synthesis process in the sample was fully completed by gas consumption. Here, the hydrate saturation was calculated by the Soave–Redlich–Kwong (SRK) equation.

(2) Permeability test of hydrate-bearing sediment before fracturing:

The pre-cool water was injected into the hydrate-bearing sediment from bottom to top. The free methane was released from the top and substituted by pre-cool water. Then the constant pressure difference between the two ends of the reactor was constantly adjusted to conduct the liquid seepage experiment. When the discharge rate is stable in the flowmeter, the average flow rate is applied to calculate the sediment-water permeability.

(3) Hydraulic fracturing test:

The proppants were added from the hopper to the proppant warehouse. After the permeability test and water displacement, the fracturing fluid with red colour was pumped into the proppant warehouse by the fracturing pump, and mixed with the proppants above the pore pressure(about 1 MPa). When the electric valve opened, the fracturing fluid and proppant mixture entered through the hydraulic jet pipe and breakthrough the thin filter paper and fractured the HBS. A camera recorded the fracturing

process in front of the sapphire cylinder.

(4) Permeability test of hydrate-bearing sediments after fracturing:

The permeability test is conducted after fracturing as (2) tests.

The experimental tests are shown in Table 1.

**Table 1 Fracturing experiment conditions and grouping**

| Test | $S_h$ (%) | Method | Fluid viscosity (MPa s) | Flow rates (ml/min) | T (K) | Porosity (%) | Loader stress (MPa) |
|---|---|---|---|---|---|---|---|




| 1 | 39.7 | Hydraulic fracturing | | | | | |
|---|------|----------------------|------|---|-----|----|----|
| 2 | 42.2 | Nitrogen foam fracturing | 2.98 | 1 | 277 | 34 | 11 |

Figure 7 shows the pressure and temperature changes during the hydrate formation process in the test. Different volumes of deionized water were added to dry sand for different hydrate saturation. The hydrate saturation was calculated by gas pressure drop via the SRK equation. The hydrate saturation of the two tests is 39.7% and 42.2%, respectively. Two different fracturing methods, namely nitrogen foam fracturing group and hydraulic fracturing group, are applied.

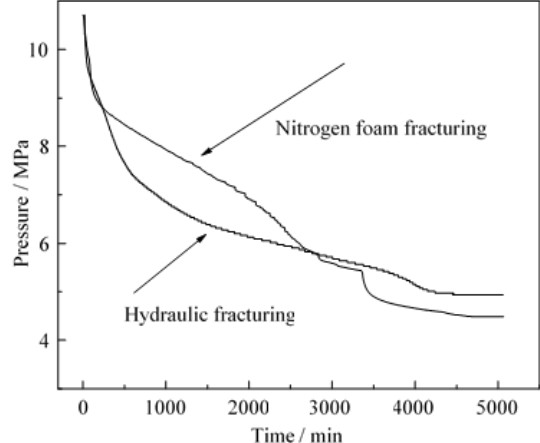

**Figure 7 Pressure and temperature curves during the formation of hydrate-bearing sediments**

**3.2 Hydraulic fracture test in hydrate bearing-sediments**

After hydraulic fracturing in Test 1, there is no more obvious ductile fractures were photographed around the sapphire reservoir. Therefore, the pressure changed little but did not significant climbing during hydraulic fracturing.

In Test 2, the gas fracturing group can be supplied a high enough pressure in the fracturing fluid to fracture the HBS at a
guaranteed flow rate. As shown in Figure 8, the fracture open and closure can be seen from the sapphire windows. The expansion of fracture is from 0 mm to 0.96 mm and then reduced from 0.96 mm to 0.58 mm in 1 min. Figure 9 shows the changes in pressure and temperature in the reactor before and after fracturing. The fracture pressure of the HBS at this point is 14.42 MPa, and the extension pressure of the fracture reached 9.54 MPa. Figure 10 shows the changes in axial stress and sediment subsidence before and after the instant of fracturing. The axial stress and subsidence of HBS increase to 0.51 MPa
and 0.53 mm, respectively. Then the subsidence of HBS retreats to 0.38 mm, which corresponds to fracture closure in Figure 8.

The hydraulic fracturing experiments verified the fracture ability of HBS.



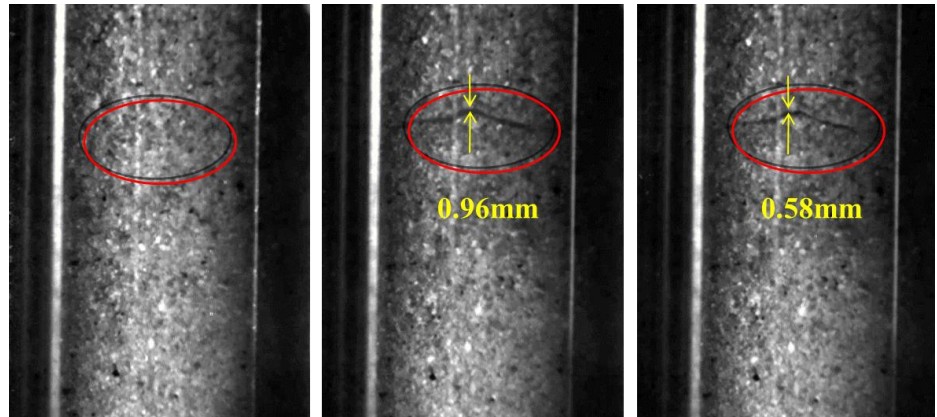

**Figure 8 Sediment nitrogen foam fracturing group burst - closure process in 1 min**

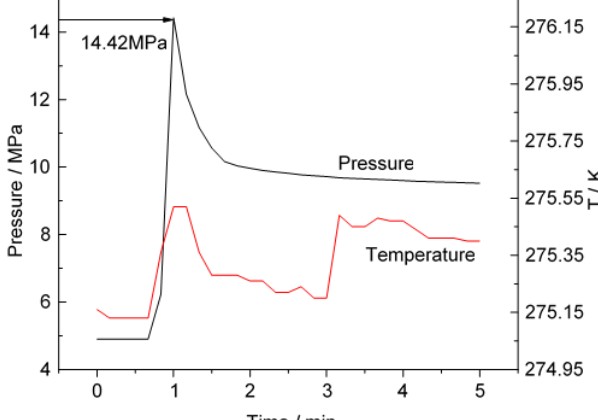


**Figure 9 Nitrogen foam fracturing group before and after fracturing instantaneous pressure and temperature changes**

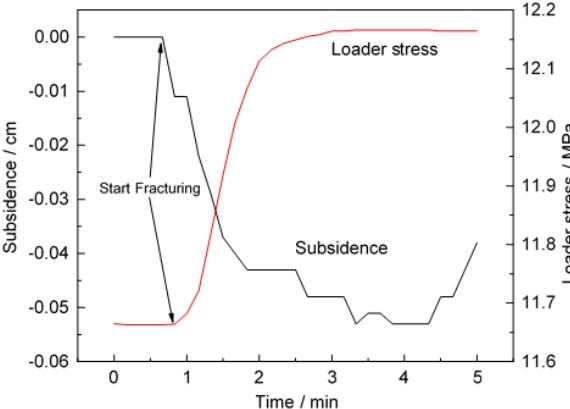

**Figure 10  Nitrogen foam fracturing group before and after fracturing instantaneous axial compression and sediment deposition**



## 4. Conclusion

The design purpose of this apparatus is to study the hydraulic fracture mechanism of hydrate exploitation and provide support for the application of reservoir reformation technology in GH reservoirs. Two pilot experiments were conducted using liquid and gas hydraulic fluids, respectively to investigate the applicability of this system.

According to previous experience, this apparatus creative developed a visualization test platform of
hydraulic fracture in HBS, with the function of movable stress, in situ GH synthesis, and deformation monitoring. The apparatus can carry out the in-situ synthesis of HBS and the tests of reservoir reformation experiments during HBS exploitation in the same environment, and provide the visual fracturing and reservoir deformation monitor. Through the pilot experiment in the early stage, the basic physical parameters of HBS fracture were collected, and the experimental steps of in-situ hydrate synthesis and
fracturing in the HBS were verified.

Furthermore, this apparatus also had well commonality and flexibility. A series of visual experiments with low temperature and high pressure, such as water jetting in HBS, and $CO_2$ hydrate geology sequestration-related experiments, are planned soon.

### Acknowledgements

This work is supported by the Science and Technology Planning Project of Guangdong Province (2021A0505030053), Natural Science Foundation of China (52004261, 52174009, 5161101020 and 51976227), Guangzhou Science and Technology Planning Project (202201010591), Guangdong Major Project of Basic and Applied Basic Research (2020B0301030003), Special project for marine economy development of Guangdong Province (GDME-2022D043), Guangdong Special Support Program
(2019BT02L278) and China Scholarship Council (202104910253).

### Data availability

The data that support the findings of this study are available from the corresponding author upon reasonable request.

### Author contribution:

JS Lu Writing - original draft, Writing - review & editing;



YX Yao: Data curation, Visualization;

DL Li: Conceptualization, Funding acquisition, Supervision, Writing - review & editing;

JH Yang: Conceptualization, Writing - review & editing;

Andy Leung: Conceptualization, Writing - review & editing;

DQ Liang: Conceptualization, Supervision;

YQ Zhang: Investigation, Resources, Visualization;

DC Lin: Visualization

**Competing interests**

The authors declare that they have no conflict of interest.

**Review statement**

This paper was edited by Rolf Müller and reviewed by Euan Nisbet and one anonymous referee.

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
