# Peer review of "A Hydrate Reservoir Renovation Device and Its Application in Nitrogen Bubble Fracturing"

_EGUsphere, 2023_

## Community Comment (CC3)

Dear Editors,

We just completed a manuscript entitled **"A Hydrate Reservoir Renovation Device and Its Application in Nitrogen Bubble Fracturing"**, now submit it to your **Geoscientific Instrumentation, Methods and Data Systems**.

No conflict of interest exists in the submission of this manuscript, and the manuscript is approved by all authors for publication. I would like to declare on behalf of my co-author that the work described was original research that has not been published previously and is not under consideration for publication elsewhere, in whole or in part. The authors listed (Jingsheng Lu, Yuanxin Yao, Dongliang Li, Jinhai Yang, Deqing Liang, Yiqun Zhang, Decai Lin and Kunlin Ma) have approved the manuscript that is enclosed.

We would like to thank the editor and the reviewers for their careful review of their manuscript. We have revised the manuscript in accordance with the reviewers' comments. The major changes are figures, introduction and results, the blue letter mark the major changes. In addition, our point-by-point response to the reviewers' comments is attached.

Further, we believe that this paper will be of interest to the readership and attract the citations of your journal.

Corresponding Author: Prof. Dongliang LI
Guangzhou Institute of Energy Conversion, Chinese Academy of Sciences
No.2, Nengyuan Rd, Tianhe District, Guangzhou, 510640, China
E-Mail: lidl@ms.giec.ac.cn

If you have any other questions, please don't hesitate to contact me.
Looking forward to your positive response.

Best regards,
Prof.LI Dongliang

**RC1**: 'Comment on egusphere-2023-1141', Anonymous Referee #1, 12 Jan 2024  reply

The authors developed an experimental apparatus which was applied to analyze the hydraulic fracture mechanism in synthesized hydrate bearing sediments. They discussed the basic principles of this apparatus and gave the preliminary experimental data. Generally speaking, the design concept of the apparatus is novel and practical. I have several questions for the authors to clarify.

1. Please check your language with a native speaker.

Thank you. We checked our language by a native speaker. The changes were marked by blue.

2. The labels of Figure 3 and Figure 4 can be more clearly.

Thank you. We updated the size of labels in Figure 3 and 4.

3. The conclusion can be slightly expanded, and include some specific future work as well. This system may be applied to CO2 geology sequestrated in the saline aquifer.

We added:This facility is also applied to $CO_2$ geology sequestrated in the saline aquifer, which can visual of the saline aquifer during the $CO_2$ injected and sequestered.

**RC2**: 'Comment on egusphere-2023-1141', Anonymous Referee #2, 24 Jan 2024  reply

This paper discussed the instruments for reservoir reformation behaviour of the hydrate-bearing sediments. Hydraulic fracturing is one of the useful stimulation technologies widely applied to the "shale gas revolution" , it is also significant to enhance production technology for gas hydrate. Overall, the manuscript is well-organized. But I also find several points and parts that need to be revised to enhance the audience reach prior to acceptance for publication. The detailed comments are as follows:

1.The language of this manuscript can be further improved to enrich the audience.

Thank you. We checked our language by a native speaker. The changes were marked by blue.

2.There are some small issues and typos (capitalization, double punctuations). Please make a thorough check.

Thank you. We checked the whoe paper. The changes were marked by blue.

3.Some legend of figures are too small, it is hard to read in paper.

Thank you. We updated the size of labels in Figure 3 and 4.

4.What is the temperature of the pre-cold water, and whether it will affect the hydrate dissociation during the permeability test?

Thank you. The temperature of the pre-cold water is 277 K. The affect of hydrate dissociation during the permeability test is not discussed in this experimental facility paper.

5.Whether the difference between the two synthesized hydrate samples will affect the properties of the reservoir?

Thank you. The heterogeneous of HBS sample will affect the properties of the reservoir,this is the common problem during HBS tests.

6.The change of hydrate reservoir permeability after hydraulic fracturing is not described in the paper?

Thank you for your comments. We added:

**3.3 Permeability test in hydrate-bearing sediments**

The permeability of hydrate-bearing sediments are tested by Darcy's law. The permeability $K$ is calculated from the flow rate q, cross-sectional area $A$, pressure differential $\Delta P$, viscosity $\mu$, and the space coordinate in the flow direction $L$. The inject pressure P1 is pumped water by the constant-flux pump, while the the outlet pressure P2 is measured. The pressure differential $\Delta P$ will decrease after the operation of hydraulic facture, so the permeability $K$ will increase in HBS.

$$K = \frac{q\mu L}{A\Delta P} \tag{1}$$

$$\Delta P = P_1 - P_2 \tag{2}$$

**RC3**: 'Comment on egusphere-2023-1141', Anonymous Referee #3, 16 Feb 2024  reply
The manuscript developed an experimental facility and conducted experiments to investigate the hydraulic fracture mechanism in synthesized HBS, which is a commendable endeavor and of significant importance for the research on methane recovery from gas hydrate reservoirs. However, a few aspects need to be addressed to improve the quality of the manuscript.

In Figure 7, the temperature curves and the corresponding axis are missing.

A: Thank you for your comments. We show the temperature in Figure 9.

The author should explain the method they use to measure the permeability of the sediments before and after hydraulic fracture. This detail is crucial for understanding the experimental procedure and interpreting the results accurately.

A: Thank you for your comments. We added:

[revised manuscript text omitted]